# The Sample Error Pre-Antimicrobial Susceptibility Testing and Its Influencing Factors from the Perspective of Hospital Management: A Cross-Sectional Study

**DOI:** 10.3390/antibiotics11121715

**Published:** 2022-11-28

**Authors:** Yuanyang Wu, Qianning Wang, Feiyang Zheng, Tiantian Yu, Yanting Wang, Si Fan, Xinping Zhang

**Affiliations:** School of Medicine and Health Management, Tongji Medical College, Huazhong University of Science and Technology, Wuhan 430030, China

**Keywords:** sample errors, antimicrobial susceptibility testing, hospital management, influencing factors

## Abstract

The overuse of antibiotics remains serious and has led to a dramatic increase in antimicrobial resistance, which poses a significant threat to global public health, although much action has been taken by World Health Organization and countries. As the direct evidence to guide the prescribing of antibiotics, the Antimicrobial Susceptibility Testing (AST) results were biased by sample errors, which was urgent and poorly explored in quality management. This study aimed to analyze sample errors pre-AST and its influencing factors from the perspective of hospital management, to provide evidence for promoting rational antibiotic use and antimicrobial stewardship. A cross-sectional survey of 5963 clinical nurses involved in pathogenic sample collection was conducted in 118 public hospitals in Hubei province, China. Dependent variables were sample errors attributed to resources and technology-oriented, capability-oriented, and attitude-oriented errors, which were measured by times per week. Independent variables were sample management information such as guidelines, record after collection, performance appraisal, training, and publicity activities, in which guidelines, record time, and record method were dummy variables, with 1 indicating having and 0 not. Others were continuous variables ranging from 0 to 4 times per month. Ordinary Least Square regression models were performed to analyze influencing factors on sample error times. The averages of sample errors on resource and technology-oriented, capability-oriented, and attitude-oriented were 1.48, 1.35, and 1.36 times per week, and their proportion were 25.3%, 38.9%, and 35.8%, respectively. The results showed that recording timeliness (r = −0.354, *p* < 0.01), record using both paper and digital methods(r = −0.060, *p* < 0.01), performance appraisal(r = −0.188, *p* < 0.01), and publicity activities (r = −0.186, *p* < 0.01) significantly reduced times of resource and technology-oriented errors. Performance appraisals(r = −0.171, *p* < 0.01) and training activities (r = −0.208, *p* < 0.01) had a positive impact on the reduction of capability-oriented error times. The times of attitude-oriented error decreased significantly when recording timeliness (r = −0.299, *p* < 0.01), implementing performance appraisal (r = −0.164, *p* < 0.01), and training activities (r = −0.188, *p* < 0.01). This study found that there was a high frequency of sample errors in quality management, especially capability-oriented and attitude-oriented errors. Sample information management, performance appraisal, training, and publicity activities were associated with the quality of samples valuable for the rational use of antibiotics.

## 1. Introduction

Irrational antibiotic use has led to a dramatic increase in antibiotic resistance (AMR), posing a significant threat to global public health. It was estimated that 1.27 million people died directly from antibiotic resistance in 2019, with 4.95 million deaths linked to antibiotic-resistant infections, and economic losses exceeding US $100 trillion [1,2].

To cope with this issue, the World Health Organization drafted the Global Action Plan on AMR in 2015, which emphasized the optimized use of antimicrobial medicines in humans and animals globally [3]. Many countries also took specific measures, such as China’s *Guidelines for the Clinical Use of Antimicrobial Drugs (2015 Edition)*, providing a strict guidance on the usage, dosage, and management of antimicrobial drugs [4]. However, the irrational use remained serious because of lacking specific prescription guidance on antibiotic use in clinical practice.

The report of Antimicrobial Susceptibility Testing (AST) was the direct evidence to guide prescribing of antibiotics, and helped physicians select the most effective antibiotic therapy [5]. However, AST results were often affected by errors in the sample collection [6]. It was estimated that approximately 80% of incorrect test reports were traced to sample quality problems [7]. Additionally, sample quality problems often included contaminated samples, tube filling errors, inappropriate containers, missing tubes, patient identification errors, et al. [8].

Many studies demonstrated that hospital management practices had positive effects on reducing sample errors and improving the accuracy of AST results. Firstly, in terms of information management, Liu et al., found that building a laboratory information management system (LIS) to monitor, count, and analyze laboratory reports was effective in reducing incorrect reports [9]. Yu et al. found that labeling errors significantly decreased after the implementation of a barcode recording system on pathology specimens [10]. Halwachs-Baumann and Winninger found that implementation of pre-labelled barcode tubes and the Greiner eHealth Technology (GeT) system in a general hospital for the exact documentation of blood collection time and the person’s name was helpful in reducing a biased result, in case the delay between sample collection and analysis was too long [11]. Secondly, Vasset found that specimen errors were associated with clinical nurses’ quality awareness, accountability, organizational commitment, and job satisfaction [12]. Sepahvand et al. found that the performance appraisal process improved the organizational commitment and job satisfaction of nurses, and significantly improved the quality of healthcare services based on Social Security Hospital [13]. Thirdly, training activities also reduced sample errors. Li et al. established a step-by-step training system for guiding staff to collect samples correctly [14]. Finally, publicity and education campaigns were also effective to reduce sample errors. Zhan and Cao found that publicity raised awareness of quality among clinical staff and avoid subjective bias [15]. Corkill et al. found that the use of educational toilet posters had a positive impact on reducing the rates of haemolysed samples and helping clinical staff to receive ongoing education [16].

In summary, hospital management practices were important for improving the sample quality pre-AST. In clinical practice, the collection of pathogenic samples was often ignored and was a weakness in quality management, often accompanied by the low investment of resources, and the lack of clinical guidance and management systems. Therefore, based on the perspective of hospital management, it was important to analyze the factors influencing the quality of the pathogenic sample pre-AST to improve the accuracy of results for promoting rational antibiotic use.

Different from previous studies that only focused on the influencing factors of the sample collection process, we described a high incidence of sample errors pre-AST and creatively divided sample errors into three categories: resource and technology, attitude and ability-oriented errors based on hospital management, and modelled times of sample errors pre-AST and its influencing factors through an Ordinary Least Square regression from a hospital management perspective. The contribution of this study was that, firstly, current research generally focused on the improvement of the AST method, such as developing a system for the rapid detection and reporting of resistant bacteria, while ignoring the sample collection errors, which was also a key loophole in quality management. Secondly, a quantitative analysis was used to categorize the sample errors in perspective of resources, capability and attitude, different from the previous classification of laboratory errors, which was valuable for selecting strategies to cope with the errors. Finally, it further enriched the study of influencing factors on sample errors from quality management, providing an evidence-based reference for hospital management practice.

## 2. Methods

### 2.1. Study Design and Participants

This was a cross-sectional design including a survey of 5963 clinical nurses in 188 public hospitals in Hubei province, China. The survey covered the demographic characteristics (gender, age, department, tenure, title, and tertiary hospital), sample errors information (site infections, wrong site selection, deficiency of equipment/technology, inappropriate containers, insufficient sample volume, samples without tagging and so on) and hospital management information (records, performance assessment, training, publicity activities, and clinical guidelines). The questionnaire items were designed with reference to *Specimen collection and transport in clinical microbiology* by the National Health Commission of the People’s Republic of China in 2018 and the *2015 edition of the Guiding Principles of Clinical Application of antibiotic*. Detailed information on the questionnaire design and data collection can be obtained at the hospital infection center of Hubei Province (whcdc. ORG).

### 2.2. Sample Errors’ Measurement

Dependent variables were designed as resource and technology-oriented errors, capability-oriented errors, and attitude-oriented errors based on six sample errors found in this survey and Anoosheh’ study. The Anoosheh’ s research objects and measurement of nurses’ behavior provided a reference for our study. As Anoosheh identified, lack of necessary equipment was an important factor resulting in clinical errors [17]. Capability-oriented errors referred to site infections and wrong site selection, which were mainly related to the skills and knowledge of clinical nurses. Attitude-oriented errors referred to insufficient sample volume and samples without tagging, which were mainly related to the lack of responsibility and awareness. Resource and technology-oriented errors included deficiency of equipment/technology and inappropriate containers during sample collection. Dependent variables as continuous variables were measured by times per week. The times of the errors were obtained from nurses’ self-report survey and were used to measure nurses’ clinical behavior, including resource and technology-oriented, ability-oriented, and attitude-oriented errors, which were different from the previous studies focusing on errors in process of patient preparation, sample collection, sample transfer, and sample analysis.

### 2.3. Hospital Management Factors’ Measurement

Recording timeliness and records methods, performance appraisal, training, publicity activities, and clinical guidelines were selected as independent variables. Recording timeliness referred to whether there was a record after collection or not, with 1 indicating having and 0 not. Records methods referred to a dummy variable with 1 indicating both using paper and digital methods to record and 0 only one of paper or digital. Clinical guidelines referred to whether the hospital had clinical guidelines for sample collection, with 1 indicating having and 0 not. Performance appraisal referred to times that nurses were appraised on the collection of samples per month within the hospital. Training referred to times that nurses were trained on the collection of samples per month. Publicity activities referred to times that nurses attended publicity activities on sample collection each month. Above continuous variables ranged from 0 to 4 in which 4 means equal to and more than 4 times per month. In addition, gender, age, tenure, title, and department were selected as control variables.

### 2.4. Statistical Analysis

Percentages and frequencies were used to describe the categorical variables. A Chi square or *t*-test was conducted for the association between influencing factors and times of sample errors. Ordinary Least Square regression was used to identify the relationship between hospital management factors and the times of sample errors. The specific model was set up as follows.
$$Y_{i}=\beta_{0}+\beta_{1}X_{1i}+\beta_{2}X_{2i}+\ldots \beta_{k}X_{ki}$$

$Y_{i}$ indicated the times of sample errors, including resource and technology-oriented, capability-oriented, or attitude-oriented errors. $X_{ki}$ denoted hospital management factors and individual demographic characteristics of clinical nurses. $\beta_{0}$ was intercept parameter and $\beta_{k}$ was the coefficient of variable, indicating the direction of the relationship between hospital management factors and times of sample errors. All statistical analyses were completed using STATA version 12.0, reporting the level of statistical significance of the coefficients (* *p* < 0.10, ** *p* < 0.05, *** *p* < 0.01).

## 3. Results

### 3.1. Characteristics of Clinical Nurses, Hospital Management Factors and Sampling Errors

The clinical nurses’ average age was 31.15, and the average tenure was about 9.41 years. There were 46.22% of nurses with primary degree title, 47.99% who worked in tertiary hospitals, the most nurses were in the Orthopedics department (16.4%) and the least in the Obstetrics and gynecology department (1.59%). The study involved 5848 female nurses (Table 1).

In terms of hospital management factors, 88.45% of clinical nurses had clinical guidelines and about 92.37% kept records after collection, with 61.29% of them keeping both paper and digital records. About 56.93% of nurses received performance appraisal at least once a month, about 79.79% attended training at least once a month, and about 69.28% attended publicity activities at least once a month (Table 1).

In terms of the distribution of sample errors, resource and technology-oriented errors were average 1.48 times per week, capability-oriented errors were average 1.35 times per week and attitude-oriented errors were average 1.36 times per week, respectively, accounting for 25.3%, 38.9%, and 35.8% of total errors (Table 2).

### 3.2. Sampling Errors and Their Influencing Factors

Chi-square or T test found that demographic characteristics, such as age, department and hospital level, and hospital management, including performance appraisal, training activities, and publicity activities were significantly associated times of sample errors. Table 3 showed the Ordinary Least Square regression results on the times of sample errors and influencing factors, in which hospital management factors as key independent variables, and demographic characteristics as control variables were included in every model.

### 3.3. Resource and Technology-Oriented Error and Its Influencing Factors

Recording timeliness after sample collection was effective in reducing resource and technology-oriented errors (r = −0.354, *p* < 0.01). Also, recording using both paper and digital methods was effective in reducing errors compared to only a paper or digital record (r = −0.060, *p* < 0.01). Performance appraisals (r = −0.188, *p* < 0.01) and publicity activities (r = −0.186, *p* < 0.01) were also effective in reducing such errors.

### 3.4. Capability-Oriented Error and Its Influencing Factors

Performance appraisals and training activities for clinical nurses were effective in reducing capability-oriented errors. Specifically, when attending four or more performance appraisals per month, the times of capability-oriented errors decreased by 0.171(r = −0.171, *p* < 0.01). Additionally, the capability-oriented error times decreased significantly when nurse staff attended three training activities per month (r = −0.208, *p* < 0.01).

### 3.5. Attitude-Oriented Error and Its Influencing Factors

Recording timeliness after the collection was effective in reducing the times of attitude-oriented errors (r = −0.299, *p* < 0.01). When attending performance appraisals (r = −0.164, *p* < 0.01) or training activities (r = −0.188, *p* < 0.01) three times a month, the attitude-oriented error times decreased significantly.

## 4. Discussion

We found that sample quality was a serious problem pre-AST, the averages of sample errors on resource and technology-oriented, capability-oriented, and attitude-oriented were 1.48, 1.35, and 1.36 times per week, and the proportion were 25.3%, 38.9%, and 35.8% respectively. Further, in hospital management, recording timeliness, recording using both paper and digital methods, performance appraisal, training, and publicity activities had a positive effect on reducing times of errors.

### 4.1. High Frequency of Sample Errors Pre-AST

Sample quality pre-AST was a serious problem with a high frequency of sample errors. Specifically, this was a first report on the average times of sample errors and attributed sample errors to resource and technology-oriented, capability-oriented, and attitude-oriented errors. The above frequency of sample errors ranged from 1.35 to 1.48 times per week implying that sample quality was urgent according to quality indicators [18]. For the proportion of three types, the capability-oriented error reached 38.9% and the attitude-oriented errors and resource and technology-oriented errors were 35.8% and 25.3%. Compared with previous findings of preanalytical mistakes ranging from 0.02% to 49.86%, the sample errors in this study were still higher than the level of quality indicators related in nearly half of the reports [18,19,20]. The previous studies only described the frequencies and classified sample errors by detection steps, such as the pre-analytical process (61.9%), the analytical process (15%), and the post-analytical process (23.1%), which paid particular attention to quality management of detection process and were technology-oriented but ignored the influencing factors that were management-oriented. We found these errors lied in lack of resources, technology, capability, and attitude from the hospital management and reported them in different types based on the causes pre-AST. Our results implied that sample quality deserved attention from hospital managers because of its high frequency of errors and the importance of hospital quality management.

### 4.2. Recording Timeliness to Reduce Resource and Technology-Oriented Errors and Attitude-Oriented Errors

The recording timeliness of the sample collection process was effective in reducing sample error times. The importance of recording was emphasized by international standards specifically for medical laboratories (ISO 15189). On the one hand, the recording collection process reduced labelling errors and improved sample quality. Yu et al. analyzed the impact of implementing a barcode recording system on the quality of pathology specimens and found a significant reduction from 15 cases to a single case after implementing recording system (*p* < 0.01) [10]. Halwachs-Baumann and Winninger found that there was a 29% reduction after implementation of pre-labelled barcode tubes and the Greiner eHealth Technology (GeT) system for exact documentation of the time of blood collection and the name of the person who did the collection, in case the delay between sample collection and analysis was too long [11]. The PDCA (Plan, Do, Check, Action) cycle management process emphasized the positive effect of a checking, collection registration, and sign-off system, which increased the accuracy of samples from 98.71% to 99.58% [21].

### 4.3. Performance Appraisal to Reduce Resource and Technology, Capability, and Attitude-Oriented Errors

Performance appraisal was effective in reducing the times of resource and technology, capability, and attitude-oriented errors. On the one hand, performance appraisal improved the motivation of nursing staff because it gave specific feedback about the need for development and helped employees to continue to excel by giving positive reinforcement, which was closely related to better performance [12]. On the other hand, performance appraisal was associated with organizational commitment and job satisfaction, for it acted as a fulcrum in continuous communication and satisfaction of employees. Sepahvand et al. based on Social Security Hospital in Iran found that the performance appraisal process raised the rate of organizational commitment and job satisfaction from 61.12 to 71.06 (*p* < 0.001), and prevented some problems such as job dissatisfaction [13]. Therefore, the establishment of the performance appraisal system had a significantly positive effect in reducing sample errors to realize quality management.

### 4.4. Training to Reduce Attitude and Capability-Oriented Errors

Training activities were effective in reducing the times of sample errors. Some common sample errors, such as clotting samples and haemolysed samples, were mainly caused by unskilled nurses due to lack of knowledge and ability on sampling. Lai et al. established a clinical laboratory quality management system to promote the communication and training of clinical nurses and found that after the implementation of training, the unqualified rate decreased from 1.29% to 0.42% (*p* < 0.01) [22]. Similarly, Li et al. established a step-by-step training system for guiding staff to collect samples correctly. The intervention of the training system resulted in a change of disqualification rate from 1.36% to 0.94% because of the improvement of nurses’ quality awareness and behavior [14]. Further, Chavan et al. provided annual training in the form of lectures for medical staff and extended these trainings to the new nurses, and there was a significant reduction (OR = 0.744, *p* < 0.01) of haemolysed specimens after interventions [23].

### 4.5. Publicity Activities to Reduce Resource and Technology-Oriented Errors and Attitude-Oriented Errors

Publicity activities had a positive effect on reducing sample error times. Publicity campaigns were one of the educational interventions, which played an important role in reducing sample errors in the clinical laboratory. On the one hand, educational interventions increased the knowledge acquisition of clinical nurses. Corkill et al. reported a 19.72% reduction in clinical haemolysed samples by using educational toilet posters to reduce the haemolysed samples and helping clinical staff to receive ongoing education (*p* < 0.001) [16]. On the other hand, educational interventions changed the clinical practice of clinical nurses. A lack of awareness that the adoption of unsuitable preanalytical procedures can generate adverse clinical, organizational, and economic consequences made the whole process more vulnerable and prone to errors. Bölenius et al. found that educational intervention was associated with nurses’ awareness and after implementing educational intervention, clinical nurses were more likely to make fewer errors [24]. Finally, publicity activities also improved the attention of hospital management on sample collection, giving clinical laboratories the necessary financial and equipment support to overcome technical and equipment dilemmas in clinical practice.

## 5. Conclusions

Based on data from a cross-sectional survey of 188 public hospitals in Hubei Province in 2021, this study analyzed mainly sample error times and influencing factors from the perspective of hospital management. The main findings showed that sample quality was a serious problem pre-AST, covering multiple dimensions including resource, capability, and attitude, and there was a high frequency of sample errors. Information management, performance appraisal, training, and publicity activities were associated with the quality of samples. It is recommended that a package of hospital management measures, including records, performance appraisal, training, and publicity activities, should be taken to improve the accuracy of AST for better guidance antibiotic prescribing. There are also some limitations in this study, such as the inability to verify causality for lack of intervention design and failure to consider the prescribing behavior based on the results of AST.

## Figures and Tables

**Table 1 antibiotics-11-01715-t001:** Clinical nurses’ demographic characteristic, hospital management factors, types of sample errors, and its univariate analysis.

Variable.	N	Mean/%	Resources/Technonlogy Oriented Errors		Capability-Oriented Errors		Attitude-Oriented Errors	
			χ/t	P	χ/t	P	χ/t	P
Demographic characteristic								
Gender			1.256	0.262	0.628	0.428	0.933	0.334
Male	115	1.93						
Female	5848	98.07						
Age	5963	31.15 ± 6.22	0.574	0.566	−2.182	**0.029**	−1.542	0.123
Tenure	5961	9.41 ± 6.68	0.173	0.863	−1.637	0.102	−1.775	0.076
Professional Title			8.922	0.063	3.632	0.458	7.316	0.120
Senior	1032	17.31						
Associate senior	2756	46.22						
Middle degree	1723	28.89						
Primary degree	152	2.55						
Other	190	3.19						
Department			39.617	**0.000**	75.106	**0.000**	80.004	**0.000**
Respiratory	643	10.78						
Urological surgical	597	10.01						
ICU	936	15.70						
Neurology	752	12.61						
Endocrinology	617	10.35						
Orthopedics	978	16.40						
Internalmedicine	389	6.52						
Surgery	362	6.07						
Pediatric	251	4.21						
Obstetrics and gynecology	95	1.59						
Other	351	5.88						
Tertiary hospital			4.012	**0.045**	0.475	0.491	2.294	0.130
Is tertiary hospital	2859	47.99						
No	3099	52.01						
Clinical guidelines			0.012	0.914	21.144	**0.000**	16.929	**0.000**
Have	5274	88.45						
Not have	689	11.55						
Sampling record			5.599	**0.018**	11.674	**0.001**	0.091	0.763
Have	5508	92.37						
Not have	455	7.63						
Record method			10.994	**0.001**	1.715	0.190	0.053	0.818
Paper or digital	2240	38.71						
Both	3547	61.29						
Performance appraisal			2.240	**0.025**	−2.061	**0.039**	0.4222	0.673
zero	2568	43.07						
one	2962	49.67						
two	137	2.30						
three	126	2.11						
four at least	170	2.85						
Training			0.395	0.693	−3.702	0.000	−1.459	0.145
zero	1205	20.21						
one	3596	60.31						
two	367	6.15						
three	395	6.62						
four at least	400	6.71						
Publicity			3.173	0.002	−6.988	0.000	−1.948	0.051
zero	1832	30.72						
one	3031	50.83						
two	325	5.45						
three	322	5.40						
four at least	453	7.60						

**Table 2 antibiotics-11-01715-t002:** Distribution of three types of sample errors and their items.

Samples Errors	N	%	Mean	SD
**Resources/technology-oriented errors**	**1908**	**25.3**	**1.48**	**1.15**
Deficiency of equipment/tech	949	12.6	1.17	0.68
Inappropriate containers	959	12.7	1.26	0.96
**Capability-oriented errors**	**2937**	**38.9**	**1.35**	**0.68**
Site infections	2017	26.7	1.11	0.39
Wrong site selection	920	12.2	1.12	0.45
**Attitude-oriented errors**	**2696**	**35.8**	**1.36**	**0.99**
Insufficient sample volume	2154	28.6	1.15	0.59
Samples without tagging	542	7.2	1.26	0.84

**Table 3 antibiotics-11-01715-t003:** Results of Ordinary Least Square regression on Sampling errors and its influencing factors.

	(1)	(2)	(3)
Variable	Resource/Tech-Oriented Error	Capability-Oriented Error	Attitude-Oriented Error
Age	−0.011 **	0.004	−0.007
	[−0.020, −0.002]	[−0.005, 0.013]	[−0.016, 0.002]
Tenure	0.006	0.002	0.004
	[−0.002, 0.014]	[−0.007, 0.011]	[−0.004, 0.013]
Title(reference: other title)	.	.	.
Senior	0.082	0.136 **	0.055
	[−0.018, 0.181]	[0.024, 0.247]	[−0.060, 0.170]
Associate senior	0.110 **	0.070	0.085
	[0.020, 0.200]	[−0.031, 0.171]	[−0.021, 0.191]
Middle degree	0.178 ***	0.050	0.153 ***
	[0.082, 0.275]	[−0.054, 0.155]	[0.039, 0.266]
Primary degree	0.075	0.024	0.106
	[−0.065, 0.214]	[−0.147, 0.195]	[−0.061, 0.274]
Department(reference:other department)	.	.	.
Respiratory	0.081 **	0.179 ***	0.225 ***
	[0.004, 0.159]	[0.091, 0.267]	[0.138, 0.313]
Urological surgical	0.316 ***	0.275 ***	0.430 ***
	[0.201, 0.431]	[0.171, 0.379]	[0.292, 0.567]
ICU	0.152 ***	0.323 ***	0.188 ***
	[0.080, 0.225]	[0.240, 0.406]	[0.112, 0.264]
Neurology	0.251 ***	0.250 ***	0.315 ***
	[0.160, 0.341]	[0.162,0.338]	[0.225, 0.404]
Endocrinology	0.206 ***	0.191 ***	0.152 ***
	[0.120, 0.293]	[0.102, 0.280]	[0.066, 0.237]
Orthopedics	0.176 ***	0.194 ***	0.202 ***
	[0.100, 0.251]	[0.113, 0.275]	[0.116, 0.288]
Internal medicine	0.175 ***	0.132 **	0.134 **
	[0.071, 0.279]	[0.017, 0.248]	[0.029, 0.240]
Surgery	0.159 **	0.168 ***	0.178 ***
	[0.018, 0.300]	[0.063, 0.273]	[0.066, 0.289]
Pediatric	0.139 **	0.161 ***	0.257 ***
	[0.030, 0.248]	[0.049, 0.274]	[0.142, 0.371]
Obstetrics and gynecology	0.239 ***	−0.079	0.212 ***
	[0.113, 0.366]	[−0.210, 0.053]	[0.078, 0.345]
Tertiary Hospital (reference:non-tertiary)			
Is tertiary	0.036	−0.052 **	−0.013
	[−0.008, 0.079]	[−0.094, −0.010]	[−0.060, 0.035]
Guidelines(reference:no)			
Have guidelines	−0.031	0.003	0.080 **
	[−0.115, 0.053]	[−0.074, 0.079]	[0.009, 0.151]
Sampling record(reference:no)			
Have record	−0.354 ***	−0.019	−0.229 ***
	[−0.524, −0.184]	[−0.123, 0.086]	[−0.339, −0.118]
record method:(reference:is paper or digital)			
Both record	−0.060 ***	−0.025	−0.011
	[−0.105, −0.014]	[−0.067, 0.018]	[−0.059, 0.036]
Performance appraisal(reference:0)	.	.	.
one	−0.027	0.016	0.028
	[−0.071, 0.017]	[−0.033, 0.064]	[−0.026, 0.083]
two	0.048	−0.058	0.024
	[−0.091, 0.188]	[−0.201, 0.086]	[−0.137, 0.185]
three	0.159	0.080	−0.164 **
	[−0.033, 0.351]	[−0.042, 0.203]	[−0.302, −0.025]
four at least	−0.188 ***	−0.171 ***	−0.237 ***
	[−0.270, −0.105]	[−0.273, −0.069]	[−0.340, −0.133]
Training(reference:0)	.	.	.
one	0.016	−0.056	0.012
	[−0.069, 0.102]	[−0.128, 0.017]	[−0.057, 0.080]
two	0.148 **	0.214 ***	0.192 ***
	[0.020, 0.277]	[0.073, 0.355]	[0.057, 0.328]
three	−0.023	−0.208 ***	−0.188 ***
	[−0.133, 0.087]	[−0.310, −0.106]	[−0.330, −0.045]
four at least	0.107 *	−0.102	−0.071
	[−0.014, 0.229]	[−0.225, 0.021]	[−0.197, 0.056]
Publicity(reference:0)	.	.	.
one	−0.022	0.090 ***	0.023
	[−0.092, 0.049]	[0.025, 0.155]	[−0.041, 0.086]
two	−0.032	0.049	0.043
	[−0.151, 0.088]	[−0.071, 0.169]	[−0.067, 0.153]
three	−0.041	0.184 ***	0.252 **
	[−0.156, 0.074]	[0.084, 0.285]	[0.027, 0.477]
four at least	−0.186 ***	0.206 ***	0.137 **
	[−0.279, −0.093]	[0.102, 0.309]	[0.024, 0.250]
_cons	0.778 ***	0.162	0.522 ***
	[0.469, 1.088]	[−0.108, 0.431]	[0.258, 0.787]
*N*	5679	5679	5679

Note: indicated the significance level, * *p* < 0.10, ** *p* < 0.05, *** *p* < 0.01.

## Data Availability

The data that support the findings of this study are available from the corresponding author (X.Z.) upon reasonable request.

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
