# Peer review of "The Sample Error Pre-Antimicrobial Susceptibility Testing and Its Influencing Factors from the Perspective of Hospital Management: A Cross-Sectional Study"

_antibiotics, 2022, doi:10.3390/antibiotics11121715_

Round 1

Reviewer 1 Report

It is an interesting article, with emphasis on identifying fhospital management practices impacting sampling errors and affecting antibiotic-susceptibility. tests.  

Under the abstract , on line 17, you introduce a study done by. Anoosheh, in Iran, without. delving into details as to how /why that would be applicable or relevant in China, where the current study has been conducted. 

Lines 32-35:Implementation sample information management, performance appraisal, training and publicity activi ties were important factors in improving the quality of samples valuable for rational use of antibi otic. 

Was there a study done AFTER implementing these measures? 

While the authors describe the study as cross sectional, in the results and in their conclusions, they state : " Information man agement, performance appraisal, training and publicity activities were important factors in improving the quality of samples. "

How did they reach this conclusion based off of their study design?

Was there different times of data collection ( before and after implementation of these ideas) that led the authors to draw their conclusions?

Or is is their supposition, that having identified these factors as the possible causes for sampling error, that it would potentially reduce the errors?

Reviewer 2 Report

Dear Editor,

The authors investigated the pre sampling errors and influencing factors. 

The title should contain the study type.

Even though it was explained in detail what is known and what this study adds should be underlined in the introduction section.

The methods and results were quite complicated to understand. The readability should be improved. A clarification should be beneficial for the audience to understand and keep reading the manuscript.

Even though this study investigated the antimicrobial-related subject, it mainly focuses on statistical and analytical investigations. I am hesitant that the journal of antibiotics is the right journal for this article. 

I believe this study is significant enough to publish in the journal of antibiotics after major issues are resolved. The aim and scope of the journal should be evaluated again.

Sincerely yours

Round 2

Reviewer 2 Report

Dear Editor,

 The authors investigated the pre-sampling errors and influencing factors. I would like to thank the authors for considering my suggestion related to their study. I believe after revision this study is significant enough published in antibiotics journal.

Regards,